# SAMPLE EFFICIENT DEEP NEUROEVOLUTION IN LOW DIMENSIONAL LATENT SPACE

## ABSTRACT

Current deep neuroevolution models are usually trained in a large parameter search space for complex learning tasks, e.g. playing video games, which needs billions of samples and thousands of search steps to obtain significant performance. This raises a question of whether we can make use of sequential data generated during evolution, encode input samples, and evolve in low dimensional parameter space with latent state input in a fast and efficient manner. Here we give an affirmative answer: we train a VAE to encode input samples, then an RNN to model environment dynamics and handle temporal information, and last evolve our low dimensional policy network in latent space. We demonstrate that this approach is surprisingly efficient: our experiments on Atari games show that within 10M frames and 30 evolution steps of training, our algorithm could achieve competitive result compared with ES, A3C, and DQN which need billions of frames.

## 1 INTRODUCTION

Training an agent that can handle difficult tasks in complex environment is always challenging for artificial intelligence. Most works for solving such problems are using reinforcement learning algorithms, e.g. learning a value function (Watkins & Dayan, 1992), or a policy function (Kakade, 2002), or both (Sutton et al., 2000). Combined with deep neural networks, the so-called deep reinforcement learning achieved great success in playing Atari games (Mnih et al., 2015), operating robots (Yahya et al., 2017), and winning human experts in challenging competitions like Go and Dota (Silver et al., 2016; OpenAI, 2018). Recently, a portion of works are trying to combine deep neural networks with evolution strategies to solve similar problems. This approach, called deep neuroevolution, achieved competitive results compared to deep reinforcement learning on Atari Games and Mujoco tasks as well in much shorter training time due to its outstanding scalability and feasibility for parallelization (Salimans et al., 2017; Such et al., 2017).

However, large-scale deep neuroevolution is both data and computation inefficient: it consumes thousands of CPU cores and needs billions of frames and thousands of generations to obtain significant results. In this work, we introduce a novel way to improve its efficiency: we train a variational autoencoder (VAE) to encode samples, and a recurrent neural network to model environment dynamics and handle temporal information; afterward, we train a two layer policy network with latent state vectors using covariance matrix adaption evolution strategy (CMA-ES).

We evaluate our sample efficient deep neuroevolution algorithm, or in short **SEDN**, on 50 Atari Games from OpenAI Gym (Brockman et al., 2016), experiment shows that our approach surpassed DQN (Mnih et al., 2013), A3C (Mnih et al., 2016), and ES (Salimans et al., 2017) methods in several Atari games within much fewer state frames, shown in Table 1. Our key findings are listed below:

- Evolution goes faster with latent state input. SEDN takes less than 30 evolution steps with only 32 children to surpass ES (Salimans et al., 2017) in several games: the latter needs thousands of children and generations.

- Training is more data efficient. Compared to ES and asynchronous RL methods which needs billions of frames to train, SEDN takes less than 10 million frames to achieve targets.

- Training is more computation efficient. On a workstation of one Intel(R) Xeon(R) CPU E5-2640 v4 @ 2.40GHz and one Titan X Pascal GPU, SEDN takes less than 4 hours to finish the evolution, compared to ES which needs a cluster to finish.

## 2 METHOD

Many real world decision making problems could be modeled as Markov Decision Process (MDP). Since proposed by Bellman (1957), it has been applied in many disciplines, e.g. robotics, economics and game playing. A Markov decision process is defined by: a set of states $\mathcal{S}$, a set of actions $\mathcal{A}$, transition probabilities defining probability over next state given current state and action $P : \mathcal{S} \times \mathcal{A} \times \mathcal{S} \to [0, 1]$, a initial state distribution $d : \mathcal{S} \to [0, 1]$, and a reward function that maps states and actions into a scalar $R : \mathcal{S} \times \mathcal{A} \to \mathbb{R}$. A finite time horizon episode $\tau$ of $T$ steps is a sequence of state-action-reward tuples: $(s_0, a_0, r_0), (s_1, a_1, r_1), \ldots, (s_t, a_t, r_t), \ldots (s_T, \emptyset, \emptyset)$ called trajectory, where $s_t \in \mathcal{S}, a_t \in \mathcal{A}, r_t = R(s_t, a_t), s_{t+1} \sim P(s_t, a_t), s_0 \sim d_0(s)$ where $s_T$ is called terminal state.

For a finite time horizon MDP, reinforcement learning (RL) algorithms learn a parameterized policy that defining probability of actions to be taken given current states $\pi_\theta : \mathcal{S} \times \mathcal{A} \to [0, 1]$, and maximize episodic reward: $\sum_{i=0}^{T-1} r_i$, or formally:

$$\underset{\theta}{\text{maximize}} \; \mathbb{E} \sum_{i=0}^{T} R(s_t, a_t) \tag{1}$$

$$\text{where } s_0 \sim d(s), a_t \sim \pi_\theta(s_t), s_{t+1} \sim P(s_t, a_t) \tag{2}$$

Above objective function takes the expectation over episodic rewards, which could be approximated by sampling $N$ trajectories from policy $\pi$ and environment dynamics $P$ and $d$. We misuse notations such that $r(\tau)$ donates episodic reward and $p(\tau)$ represents trajectory probability, then we have below reformulation:

$$\underset{\theta}{\text{maximize}} \; L(\theta) = \sum_{i=1}^{N} r(\tau_i) p(\tau_i) \tag{3}$$

$$\text{where } p(\tau) = d(s_0) \prod_{t=0}^{T-1} \pi_\theta(s_t, a_t) p(s_t, a_t, s_{t+1}), r(\tau) = \sum_{t=0}^{T-1} r_i \tag{4}$$

Vanilla policy gradient (PG) methods optimize $L(\pi_\theta)$ by taking its gradient:

$$\nabla_\theta L \approx \frac{1}{N} \sum_{i=1}^{N} r(\tau_i) \nabla_\theta \log p_\theta(\tau_i) \tag{5}$$

$$= \frac{1}{N} \sum_{i=1}^{N} r(\tau_i) \sum_{t=0}^{T_i-1} \nabla_\theta \log \pi_\theta(s_t, a_t) \tag{6}$$

and updates policy $\pi_\theta$ to maximize expected episodic reward. Variants of policy gradient methods achieved great success in recent years, e.g. TRPO (Schulman et al., 2015) and PPO Schulman et al. (2017).

Evolution strategies (ES), instead, take another approach to solve above optimization problem without any differentiation: it approximates the gradient by sampling, shown in Equation 7.

$$\nabla_\theta L \approx \frac{1}{\sigma^2} \mathbb{E}_{\hat{\theta} \sim \mathcal{N}(0, \sigma^2)} (L(\theta + \hat{\theta}) \hat{\theta}) \tag{7}$$

Vanilla ES solves MDP problems iteratively shown in Algorithm 1: firstly, it samples Gaussian noise from a normal distribution; secondly, it injects noise into current policy, and evaluates it in environment, obtaining episodic rewards; thirdly, it updates current policy parameters by the product of episode reward and noise injected. ES and PG are interpreted as two faces of Gaussian smoothing on policy: ES is on parameter space smoothing and PG is on action space smoothing (Salimans et al., 2017).

---

**Algorithm 1** Evolution Strategies

---

**Input:** Learning rate $\eta$, noise standard derivation $\sigma$, initial policy parameter $\theta$
 1: **for** $t = 0, 1, \ldots$ **do**
 2:   Sample $N$ parameter noise: $\hat{\theta}_i \sim \mathcal{N}(\mathbf{0}, \sigma^2 \mathbf{I})$ for $i = 1, \ldots, N$
 3:   Execute perturbated deterministic policies $\pi_{\theta_i} : a_t = \arg\max_a \pi_{\theta_i}(s_t)$ to produce trajectory $\tau_i$, where $\theta_i = \theta + \hat{\theta}_i$ for $i = 1, \ldots, N$
 4:   Update policy parameter $\theta \leftarrow \theta + \alpha/(N\sigma^2) \sum_{i=1}^{N} r(\tau_i)\hat{\theta}_i$
 5: **end for**

---

ES enjoys several advantages over PG. Firstly, it is a black-box optimization method that does not need gradients. In problems where gradients are not available, or state encoding is not feasible, or gradient is blocked by some operations in neural networks, ES exhibits its wider range of applications over PG. Secondly, in low dimensional optimization problems like Mujoco tasks, ES has competitive performance with PG (Salimans et al., 2017), but is less sensitive to hyperparameters (Heidrich-Meisner & Igel, 2008). Thirdly, the exploration-exploitation trade-off needs to be balanced in RL here is integrated: policy parameter perturbation ensures different behaviors during evaluation, whereas RL has to use a stochastic policy to explore. Lastly, ES is highly scalable therefore can be trained faster by a group of workers in parallel.

However, in $n$-dimensional search space where $n$ is large, the exploration directions $O(2^n)$ increases in orders with $n$, therefore random sampling for gradient approximation would be very inefficient. For video games like Atari, high dimensional pixel state space induces more parameters of policy to be optimized, hence less efficient for ES. Surprisingly, Salimans et al. (2017) shows that even natural evolution strategies (NES) can beat common RL algorithms in several Atari games. Among the class of evolution strategies, the covariance matrix adaption evolution strategy (Hansen & Ostermeier, 2001) is the most successful one in solving low dimensional optimization problems. Would covariance matrix adaption evolution strategy (CMA-ES) perform better or faster over NES for high dimensional tasks? To answer this question, one problem has to be solved, that the application of CMA-ES on high dimensional space is infeasible since it needs to compute a large covariance matrix of size $O(n^2)$. Recently advances in deep learning show promises to encode high dimensional spatial-temporal information into low dimensional latent vectors (Oh et al., 2015; Pathak et al., 2018; Ha & Schmidhuber, 2018) by modeling environment dynamics. In this case, we can convert high dimensional RL tasks into low dimensional tasks, where CMA-ES can take its advantage.

## 2.1 COVARIANCE MATRIX ADAPTION EVOLUTION STRATEGY

CMA-ES improves ES in two ways. Firstly, vanilla ES samples noise from a fixed Gaussian distribution $\mathcal{N}(\mathbf{0}, \sigma^2 \mathbf{I})$ of only one degree of freedom $\sigma$, the step size; whereas CMA-ES, samples noise from $\mathcal{N}(\mathbf{0}, \mathbf{C})$ where covariance matrix $\mathbf{C}$ has $n(n+1)/2$ degree of freedom. Secondly, CMA-ES updates sampling parameters $\sigma, \mathbf{C}$ by rank based method, rather than updating policy parameter by approximating its gradients.

We now explain rank-based ES. Rank-based ES is a selection-recombination process. Let $f$ donates the objective function, $\lambda$ be population size, $\mu$ be parent population size, $\{w_i | w_1 > w_2 > \cdots > w_\mu > 0, \sum_{i=1}^{\mu} w_\mu = 1, i = 1, \ldots, \mu\}$ be weight coefficient for recombination. Firstly, Rank-based ES first samples $\lambda$ points from Gaussian distribution $\mathbf{x}_i \sim \mathbf{m} + \sigma \mathcal{N}_i(\mathbf{0}, \mathbf{C})$ for $i = 1, \ldots, \lambda$. Next, it evaluates sampled points on the objective function, and sort according to the cost such that $f(\mathbf{x}_1) < f(\mathbf{x}_2) < \cdots < f(\mathbf{x}_\lambda)$. In the following, it recombines to get new mean and covariance matrix $\mathbf{m} \leftarrow \sum_{i=1}^{\mu} w_i \mathbf{x}_i$, $\mathbf{C} \leftarrow \sum_{i=1}^{\mu} w_i(\mathbf{x}_i - \mathbf{m})(\mathbf{x}_i - \mathbf{m})^T$. This procedure repeats for $G$ generations.

Similar to gradient based optimization methods, CMA-ES further uses momentum, step size control and path accumulation for more stable optimization. Let $\mathbf{y}_i^{(g+1)} \sim \mathcal{N}(\mathbf{0}, \mathbf{C}^{(g)}), i = 1, \ldots, \lambda$ be sampled noise to be injected, $\mathbf{x}_i = \mathbf{m} + \sigma \mathbf{y_i}$, suppose $\{\mathbf{y}_i | i = 1, \ldots, \mu\}$ are sorted such that $f(\mathbf{x}_1) < f(\mathbf{x}_2) < \cdots < f(\mathbf{x}_\lambda)$, updating of covariance matrix with momentum gives:

$$\mathbf{C}^{(g+1)} = (1 - c_\mu)\mathbf{C}^{(g)} + c_\mu \sum_{i=1}^{\mu} w_i \mathbf{y}_i^{(g+1)} \mathbf{y}_i^{(g+1)T} \tag{8}$$

where $g$ is the number of generation and $c_\mu$ is momentum constant. In favor to generate best point from the current generation, it further adds the best point's covariance term into Equation 8:

$$\mathbf{C}^{(g+1)} = (1 - c_\mu - c_1)\mathbf{C}^{(g)} + c_1\mathbf{y}_1^{(g+1)}\mathbf{y}_1^{(g+1)T} + c_\mu \sum_{i=1}^{\mu} w_i\mathbf{y}_i^{(g+1)}\mathbf{y}_i^{(g+1)T} \tag{9}$$

In practice, exponential smoothing over parents' mean path, or evolution path, is used:

$$\mathbf{p}_c^{(g+1)} = (1 - c_c)\mathbf{p}_c^{(g)} + c_c(\mathbf{m}^{(g+1)} - \mathbf{m}^{(g)})/\sigma^{(g)} \tag{10}$$

$$\mathbf{C}^{(g+1)} = (1 - c_\mu - c_1)\mathbf{C}^{(g)} + c_1\mathbf{p}_c^{(g+1)}\mathbf{p}_c^{(g+1)T} + c_\mu \sum_{i=1}^{\mu} w_i\mathbf{y}_i^{(g+1)}\mathbf{y}_i^{(g+1)T} \tag{11}$$

Next, we explain how to control the step size $\sigma$. The step size is related to the evolution path: it should be decreased if the path is short and increased if the path is long. In contrast to evolution path $\mathbf{p}_c$, here we need to correct its directions by taking conjugate evolution path $\mathbf{p}_\sigma$:

$$\mathbf{p}_\sigma^{(g+1)} = (1 - c_\sigma)\mathbf{p}_\sigma^{(g)} + c_\sigma \mathbf{C}^{(g),-\frac{1}{2}}(\mathbf{m}^{(g+1)} - \mathbf{m}^{(g)})/\sigma^{(g)} \tag{12}$$

To update step size, we compare conjugate evolution path with the expectation of length of vectors sampled from normal distribution $\mathcal{N}(\mathbf{0}, \mathbf{I})$:

$$\ln \sigma^{(g+1)} = \ln \sigma^{(g)} + \frac{c_\sigma}{d_\sigma}\left(\frac{\|\mathbf{p}_\sigma\|}{\mathbb{E}\|\mathcal{N}(\mathbf{0}, \mathbf{1})\|} - 1\right) \tag{13}$$

Hansen (2016) suggests default hyperparameters for CMA-ES, we summarize complete CMA-ES in Algorithm 2.

---

**Algorithm 2** CMA-ES

---

**Input:** $\mathbf{m} \in \mathbb{R}^n, \sigma \in \mathbb{R}_+, \lambda, \mu \in \mathbb{N}, \mu < \lambda, \mathbf{C} = \mathbf{I}, \mathbf{p_c} = \mathbf{0}, \mathbf{p_\sigma} = \mathbf{0}$
 $w_i$ such that $\mu_w = 1/\sum_{i=1}^{\mu} w_i^2 \approx 0.3\lambda, w_1 \geq \cdots \geq w_\mu > 0, \sum_{i=1}^{\mu} w_i = 1$
 $c_c, \approx 4/n, c_\sigma \approx 4/n, c_1 \approx 2/n^2, c_\mu \approx \mu_w/n^2, c_1 + c_\mu \leq 1, d_\sigma \approx 1 + \sqrt{\mu_w/n}$

1: **for** each generation step **do**
2:  Sampling: $\mathbf{y}_i \sim \mathcal{N}_i(\mathbf{0}, \mathbf{C}), \mathbf{x}_i = \mathbf{m} + \sigma\mathbf{y}_i$ for $i = 1, \ldots, \lambda$
3:  Evaluation: obtain $f(\mathbf{x}_i)$, sort $\{\mathbf{x}_i | i = 1, \ldots, \lambda\}$ points such that $f(\mathbf{x}_1) \leq \cdots \leq f(\mathbf{x}_\lambda)$
4:  Mean update: $\mathbf{m} \leftarrow \sum_{i=1}^{\mu} w_i\mathbf{x}_i, \mathbf{y}_w = \sum_{i=1}^{\mu} w_i\mathbf{y}_i$
5:  Evolution path update: $\mathbf{p_c} \leftarrow (1 - c_c)\mathbf{p_c} + \mathbf{y}_w\sqrt{1 - (1 - c_c)^2}\sqrt{\mu_w}$
6:  Conjugate evolution path update: $\mathbf{p_\sigma} \leftarrow (1 - c_\sigma)\mathbf{p_\sigma} + \mathbf{C}^{-1}\mathbf{y}_w\sqrt{1 - (1 - c_\sigma)^2}\sqrt{\mu_w}$
7:  Covariance matrix update: $\mathbf{C} \leftarrow (1 - c_1 - c_\mu)\mathbf{C} + c_1\mathbf{p_c}\mathbf{p_c}^T + c_\mu \sum_{i=1}^{\mu} w_i\mathbf{y}_i\mathbf{y}_i^T$
8:  Step size update: $\sigma \leftarrow \sigma \exp(\frac{c_\sigma}{d_\sigma}(\frac{\|\mathbf{p_\sigma}\|}{E\|\mathcal{N}(\mathbf{0},\mathbf{1})\|} - 1))$
9: **end for**

---

CMA-ES addresses typical problems in non-linear optimization problems like ill-conditioning and ruggedness. The covariance matrix approximates the inverse Hessian matrix, and the updates approximate natural gradient by adapting the search metric into a sphere, hence increasing performance by orders of magnitude. Rank based selection ensures its invariance against translation and rotation in search space. The step size control facilitates fast convergence. However, it is not applicable to deep neural networks used in DRL which has millions of parameters and induces $O(n^2)$ order of the size of the covariance matrix. This leads to an intuitive approach by encoding frames into small latent vectors to model the dynamics of the game environment, and evolve with a lower dimensional policy parameter space.

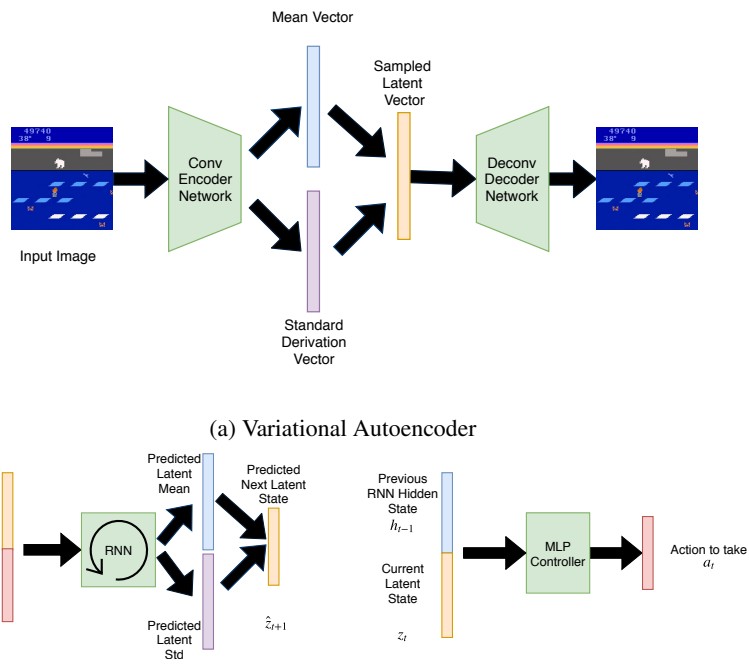

(a) Variational Autoencoder

(b) LSTM Forward Model Dynamics
(c) Policy Network

Figure 1: Our SEDN model. VAE encodes state frame into latent vector; LSTM encodes current latent state and action, predicts next latent state; policy network takes current latent state and hidden state from previous timestep of RNN as input, predict action to execute

## 2.2 ENVIRONMENT DYNAMICS MODELING FOR EPISODIC DATA ENCODING

How to learn a model that can encode episode data $\tau : s_0, a_0, s_1, a_1, \ldots s_T, a_T$? We develop a method by modeling environment dynamics. Perhaps the earliest work on building predictive model for vision based RL tasks was introduced by Schmidhuber & Huber (1991), which proposed using neural networks to predict attention region given previous frames and actions. After the success of deep learning, Oh et al. (2015) proposed an Encoding-Transformation-Decoding framework for action conditional video prediction, and applied on Atari games. More recently, Pathak et al. (2018) learns forward and inverse model dynamics to imitate expert's behavior given only image sequences. Ha & Schmidhuber (2018) also learned forward model dynamics using random policy demonstrations, and train a controller totally *inside the dream*.

In this work, we take similar approach to encode episodic data. Specifically, we first train a Variational Autoencoder (VAE) that encode spatial information; then train an recurrent neural network (RNN) to encode temporal information by learning model dynamics. Let $\phi_e : \mathcal{S} \to \mathbb{R}^n, \phi_d : \mathbb{R}^n \to \mathcal{S}$ donates our encoder and decoder network where $n$ is our latent space dimension, $f : \mathbb{R}^n \times \mathcal{A} \to \mathbb{R}^n$ be the our RNN, and $c : \mathbb{R}^n \times \mathbb{R}^m \to \mathcal{A}$ be our policy network where $m$ is the dimension of our RNN's hidden state. The encoder $\phi_e$ encodes state frame $s_t$ into a latent vector $z_t$:

$$\mu(s_t), \sigma(s_t) = \phi_e(s_t) \tag{14}$$
$$z_t \sim \mathcal{N}(\mu(x_t), \sigma(x_t)) \tag{15}$$

The decoder $\phi_d$ decodes latent vector back to state frame $\hat{s}_t$. VAE is trained to minimize the reconstruction loss between $s_t, \hat{s}_t$, and KL-divergence between latent vector and $\mathcal{N}(0, I)$:

$$\hat{s}_t = \phi_d(z_t) \tag{16}$$

$$L_{\text{vae}} = \|s_t - \hat{s}_t\|^2 + \mathcal{KL}[\mathcal{N}(\mu(s_t), \sigma(s_t))\|\mathcal{N}(0, I)] \tag{17}$$

LSTM $f$ models the environment dynamics in latent state space, and encode temporal information:

$$\mu(\hat{s}_{t+1}), \sigma(\hat{s}_{t+1}) = f(z_t, a_t) \tag{18}$$

$$\hat{z}_{t+1} \sim \mathcal{N}(\mu(\hat{s}_{t+1}), \sigma(\hat{s}_{t+1})) \tag{19}$$

LSTM is trained to minimize the KL-divergence between actual latent state from vae's encoder and predicted latent state:

$$\mu(s_{t+1}), \sigma(s_{t+1}) = \phi_e(s_{t+1}) \tag{20}$$

$$L_{\text{rnn}} = \mathcal{KL}[\mathcal{N}(\mu(\hat{s}_{t+1}), \sigma(\hat{s}_{t+1}))\|\mathcal{N}(\mu(s_{t+1}), \sigma(s_{t+1}))] \tag{21}$$

The policy network takes the latent vector of the current state, and previous timestep's hidden state of RNN, we use deterministic policy in this work:

$$a_t \sim c(z_t, h_{t-1})$$

We illustrate our model in Figure 1.

## 3 EXPERIMENTS

Our complete training pipeline is summarized in Algorithm 3. We use OpenMPI's master-slave mode (Scarabello & Clipp, 2018) for parallel training of policy network during evolution steps, Hansen (2018) for CMA-ES optimizer, and Pytorch (Paszke et al., 2017) deep learning platform to train VAE and RNN. We train the three modules separately. Firstly, we collect episode data during training the policy network using CMA-ES. Secondly, we extract all video frames from collected data and train our variational autoencoder. Thirdly, we convert all state frames into latent state vectors using VAE trained. Lastly, we train our LSTM using the converted episode data. This training cycle iterates till the end.

During the training, we set our training data buffer to be a FIFO queue of size $1M$: this will ensure the safety of workstation memory. The training process will stop when the maximum number of frames is reached or the maximum evolution step is reached, whichever is earlier. Our parallel training adopts a task-queue master-slave mode: the master node immediately distributes remaining tasks once there is any slave node available, and slave node requests new task right after it finishes a previous task. Unlike Salimans et al. (2017), our master-slave communication frequency is so low that it only takes a small percentage of time compared to episode data sampling steps.

We conduct experiments on 50 Atari 2600 games in OpenAI Gym (Brockman et al., 2016). We choose a larger encoder convolutional network compared to Vanilla DQN (Mnih et al., 2013) for better VAE performance: our experiment shows poor decoding performance if following Vanilla DQN's convolutional network. Our data pre-possessing is also different from Vanilla DQN: we resize image frames directly without frame stack, grayscale conversion or frame skip. We tried to stack latent state vector as input for policy network but found performance drop: the reason may be that increased dimension required for policy network slows down the evolution process for CMA-ES optimizer.

Our encoder network consists of 4 convolutional layers with 32, 64, 128, 256 channels followed by a flatten layer and a linear layer of 128 units. The convolutional layers use $4 \times 4, 4 \times 4, 3 \times 3, 3 \times 3$ filter size and strides of $2, 2, 1, 1$. Our decoder network reverses the operations except the last deconvolutional layer uses $6 \times 6$ filter size. Each convolutional and deconvolutional layer uses rectified activation, except that last convolutional layer uses tanh activation. Image frames are resized into $(84, 84, 3)$ to feed into our network. We set our latent state dimension to be 128. We use one layer LSTM of 128 hidden size to model environment dynamics, therefore our input dimension of policy network is 256. Our policy network is a two-layer MLP of size $256 - 32 - n_a$, where $n_a$ is the action space dimension.

The training of VAE and RNN is triggered every 8 evolution step. Each child policy is evaluated for 16 episodes, and the average episode reward is returned for CMA-ES optimizer to update. To visualize the training process of policy network, we make a boxplot of average episode reward distribution of Frostbite, shown in Figure 2 The maximum reward performance among policy networks increases significantly right after the training of VAE and RNN triggered at 8th, 16th generations. This agrees with our assumption that VAE and RNN encode spatial-temporal information from past experience and provides good feature representation for policy to evolve. It is worth to notice that for some game, e.g. Private Eye, policy performance increases significantly even without updating VAE and RNN; this observation agrees with Blundell et al. (2016) where features are extracted from random matrix projection.

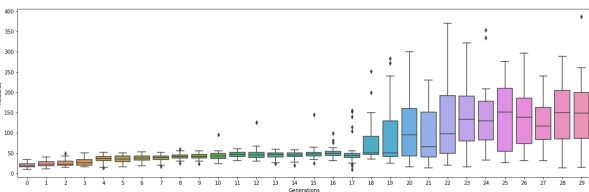

Figure 2: Boxplot of the training process of Frostbite. Maximum reward performance among policy networks increases significantly after training of VAE and RNN triggered at 8th, 16th generations.

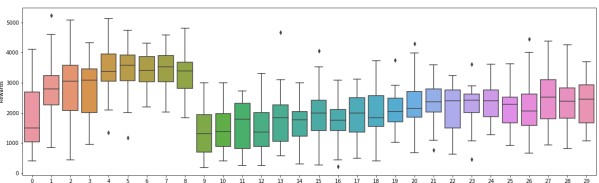

Figure 3: Boxplot of the training process of Atlantis. Maximum reward performance among policy networks drops significantly after training of VAE and RNN triggered at 8th generations.

However, we also notice that for some games, policy networks performance drops significantly right after updating of VAE and RNN, e.g Atlantis shown in Figure 3. We find that the cause may be the poor encoding performance of VAE. Shown in Figure 4, we compare VAE's encoding-decoding on Frostbite and Atlantis. We find that image decoded from VAE will ignore tiny moving objects, leaving only a static background; this issue occurs especially in rare frames far from episode starts; similar phenomenon was found in games like Kangaroo, Gravitar and Breakout.

Experiment result is summarized in Table 1. Our SEDN surpasses ES, DQN, and A3C in **7** of 50 Atari games, but with only **10M** frames, whereas DQN using 200M frames, and ES and A3C using 1B frames. Due to the limitation of time and resources, we did not conduct hyperparameter tuning on SEDN. We believe better encoding techniques to address above issues will give a boost of performance on SEDN. We left this as an open direction in future works.

## 4 DISCUSSION

Whereas traditional reinforcement learning algorithms need to balance between exploration and exploitation (Sutton et al., 1998), and even add perturbation on parameters to ensure good exploration (Plappert et al., 2017); evolution strategy directly drives exploration by sampling from optimizer's distribution parameters (Salimans et al., 2017) during evolution: it integrates exploration and exploitation. This is what makes it so efficient especially in some low dimensional control tasks like CartPole. By encoding spatial-temporal information from high dimensional pixel space into low dimensional latent space, a challenging RL task could hence be converted into a simpler task where evolution strategies have advantages over deep reinforcement learning methods.

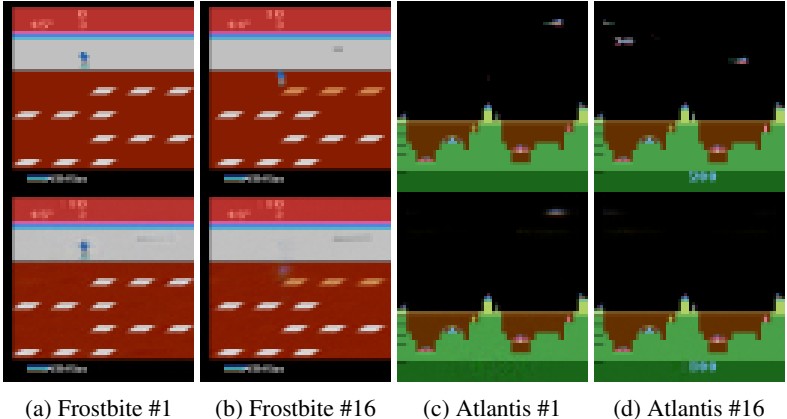

(a) Frostbite #1    (b) Frostbite #16    (c) Atlantis #1    (d) Atlantis #16

Figure 4: Comparison of VAE's encoding-decoding on Atlantis and Frostbite. Upper: Original image; Lower: decoded image from the latent vector. Notice for Frostbite, decoded image agrees with original image at the 16th frame; but for Atlantis decoder does not recover the original small objects at 16th frame

Similar to what Salimans et al. (2017) are facing: ES does not perform well on some easy games like Enduro and Breakout. One possible reason is that ES's optimized policy is close to where it is randomly initialized due to evolution path control under ES, whereas optimal policy for dense reward games like Breakout are far away from where it is randomly initialized. Another reason is that, long episodic time horizon games like Enduro are very inefficient for ES to solve: given limited number of frames, a longer episode will result in less number of episodes, hence less and slower updates for ES. In addition, episode rewards ignore temporal reward information over time steps. This issue is critical in games like Enduro. Enduro is a car racing game, in which agent obtain rewards after passing other cars and penalty if it is passed by other cars, final ranking will be accounted as the episodic reward after a long episode. A random policy will always obtain a zero episodic reward under this case, hence ES learns nothing from such games. These two issues limit the application of vanilla ES and its variants, remaining to be resolved in future works. A possible direction is a mixture of ES and RL, e.g. using ES as a warm start of RL policy.

## 5 CONCLUSION

Our work introduces a fast and efficient evolution algorithm that trains deep neural networks to play Atari games. Our experiment shows that encoding spatial-temporal information from high dimensional pixel space into low dimensional latent space makes ES fast, effective and efficient: SEDN outperforms DQN, A3C, and ES in 30 evolution steps with 10M frames experience in several Atari games. We conclude SEDN is a competitive approach to solving challenging RL tasks.

In future works, we plan to address issues discussed in section 4 by applying state of the art computer vision techniques like self-attention maps for better information encoding; we would also like to apply information encoding with RL to understand feature importance for RL.

Table 1: Result is obtained using SEDN on Atari games with 16 re-runs with up to 30 random initial no-ops compared with DQN Mnih et al. (2013), A3C Mnih et al. (2016), and ES Salimans et al. (2017); DQN, A3C, ES data are obtained from Salimans et al. (2017). Enduro's data is missing due to its long episode problem discussed in Section 4

|  | DQN | A3C, FF | ES, FF | SEDN, Ours |
|---|---|---|---|---|
| Frames | 200 M | 1 B | 1 B | **10 M** |
| Time | 7-10 day | 4 day | **4 hour** | **4 hour** |
| Amidar | 133.4 | **283.9** | 112 | 33.4 |
| Assault | 3332.3 | **3746.1** | 1673.9 | 175.9 |
| Asterix | 124.5 | **6723** | 1440 | 387.5 |
| Asteroids | 697.1 | **3009.4** | 1562 | 519.4 |
| Atlantis | 76108 | 772392 | **1267410** | 5237.5 |
| Bank Heist | 176.3 | **946** | 225 | 13.1 |
| Battle Zone | **17560** | 11340 | 16600 | 3625 |
| Beam Rider | 8672.4 | **13235.9** | 744 | 332.8 |
| Berzerk |  | **1433.4** | 686 | 224.2 |
| Bowling | 41.2 | 36.2 | 30 | **97.6** |
| Boxing | 25.8 | 33.7 | 49.8 | **55.3** |
| Breakout | 303.9 | **551.6** | 9.5 | 8.8 |
| Centipede | 3773.1 | 3306.5 | **7783.9** | 3586.9 |
| Chopper Command | 3046 | **4669** | 3710 | 3718.8 |
| Crazy Climber | 50992 | **101624** | 26430 | 23662.5 |
| Demon Attack | 12835.2 | **84997.5** | 1166.5 | 1812.4 |
| Double Dunk | **21.6** | 0.1 | 0.2 | -8.8 |
| Enduro | **475.6** | 82.2 | 95 |  |
| Fishing Derby | 2.3 | 13.6 | **49** | -75.4 |
| Freeway | 25.8 | 0.1 | **31** | 23.8 |
| Frostbite | 157.4 | 180.1 | 370 | **387.5** |
| Gopher | 2731.8 | **8442.8** | 582 | 242.5 |
| Gravitar | 216.5 | 269.5 | **805** | 131.2 |
| Ice Hockey | 3.8 | **4.7** | 4.1 | -2.9 |
| Kangaroo | 2696 | 106 | **11200** | 262.5 |
| Krull | 3864 | 8066.6 | **8647.2** | 1238.2 |
| Montezumas Revenge | 50 | **53** | 0 | 0 |
| Name This Game | 5439.9 | **5614** | 4503 | 5403.1 |
| Phoenix |  | **28181.8** | 4041 | 1258.1 |
| Pit Fall |  | **123** | 0 | 0 |
| Pong | 16.2 | 11.4 | **21** | **21** |
| Private Eye | 298.2 | 194.4 | 100 | **6674.6** |
| Q*Bert | 4589.8 | **13752.3** | 147.5 | 160.9 |
| River Raid | 4065.3 | **10001.2** | 5009 | 531 |
| Road Runner | 9264 | **31769** | 16590 | 26430.8 |
| Robotank | **58.5** | 2.3 | 11.9 | 4 |
| Seaquest | **2793.9** | 2300.2 | 1390 | 128.8 |
| Skiing |  | 13700 | 15442.5 | **16948.4** |
| Solaris |  | 1884.8 | 2090 | **2832.5** |
| Space Invaders | 1449.7 | **2214.7** | 678.5 | 148.8 |
| Star Gunner | 34081 | **64393** | 1470 | 337.5 |
| Tennis | 2.3 | **10.2** | 4.5 | -2.1 |
| Time Pilot | 5640 | **5825** | 4970 | 1518.8 |
| Tutankham | 32.4 | 26.1 | **130.3** | 12.5 |
| Up and Down | 3311.3 | 54525.4 | 67974 | 8031.8 |
| Venture | 54 | 19 | **760** | 12.5 |
| Video Pinball | 20228.1 | **185852.6** | 22834.8 | 23058.4 |
| Wizard of Wor | 246 | **5278** | 3480 | 606.2 |
| Yars Revenge |  | 7270.8 | **16401.7** | 2953.7 |
| Zaxxon | 831 | 2659 | **6380** | 262.5 |

**Algorithm 3** Complete Training Pipeline

**Input:** VAE encoder $\phi_{enc} : x_t \to z_t$, VAE decoder $\phi_{dec} : z_t \to x_t$
RNN $f : (z_t, a_t) \to z_{t+1}$, Policy $p : (z_t, h_{t-1}) \to a_t$ and CMA-ES optimizer: $g$
1: **for** each iteration **do**
2:     training data buffer $\mathcal{X} = \{\}$
    *// Train policy network and collect data*
3:     **for** each evolution step **do**
4:         sample $\lambda$ children $\{c_1, \ldots, c_\lambda\}$ from CMA-ES optimizer $g$
5:         **for** child $i$ from 1 to $\lambda$ **do**
6:             contrust policy $p_i$ from $c_i$
7:             **for** each episode $j$ from 1 to $N$ **do**
8:                 sample episode data $\mathcal{Y}_{ij} = (s_0, a_0, \ldots, s_T, a_T)$ and episode reward $R_{ij}$ with VAE, RNN and Policy:$\phi_{enc}, f, p_i$
9:                 append episode data to training data $\mathcal{X} \leftarrow \mathcal{X} \cup \{\mathcal{Y}_{ij}\}$
10:             **end for**
11:             Calculate average return for child $i$: $\bar{R}_i = \sum_{j=1}^{N} R_{ij}/N$
12:         **end for**
13:         Update CMA-ES optimizer $g \leftarrow g(c_1, \bar{R}_1, c_2, \bar{R}_2, \ldots, c_\lambda, \bar{R}_\lambda)$
14:     **end for**
    *// Train VAE and extract frames*
15:     Training image frames $\mathcal{I} = \{\}$
16:     **for** each $\mathcal{Y}$ in $\mathcal{X}$ **do**
17:         $(s_1, a_1, \ldots, s_T, a_T) \leftarrow \mathcal{Y}$
18:         $\mathcal{I} \leftarrow \mathcal{I} \cup \{s_1, s_2, \ldots, s_T\}$
19:     **end for**
20:     Train and update VAE $\phi_{enc}, \phi_{dec}$ with $\mathcal{I}$
21:     Extracted episode data $\mathcal{S} = \{\}$
22:     **for** each $\mathcal{Y}$ in $\mathcal{X}$ **do**
23:         $(s_1, a_1, \ldots, s_T, a_T) \leftarrow \mathcal{Y}$
24:         $\mathcal{S} \leftarrow \mathcal{S} \cup \{\phi_{enc}(s_1), a_1, \ldots, \phi_{enc}(s_T), a_T\}$
25:     **end for**
    *// Train RNN*
26:     **for** each $\mathcal{Y}$ in $\mathcal{S}$ **do**
27:         $(\phi_{enc}(s_1), a_1, \ldots, \phi_{enc}(s_T), a_T) \leftarrow \mathcal{Y}$
28:         $\mu_{1:T}, \sigma_{1:T} = \phi_{enc}(s_{1:T}), z_{1:T} \sim \mathcal{N}(\mu_{1:T}, \sigma_{1:T})$
29:         $\hat{\mu}_{2:T}, \hat{\sigma}_{2:T} = f(z_{1:T-1}, a_{1:T-1})$
30:         Calculate KL-divergence Loss $L = \mathcal{KL}[\mathcal{N}(\mu(\hat{s}_{t+1}), \sigma(\hat{s}_{t+1}))\|\mathcal{N}(\mu(s_{t+1}), \sigma(s_{t+1}))]$
31:         Backward NLL loss and update RNN $f$
32:     **end for**
33: **end for**
34: Construct final policy $p_f$ from CMA-ES optimizer's favorite solution
35: **return** $\phi_{dec}, \phi_{enc}, f, p_f$

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
