# OpenReview forum: "Sample Efficient Deep Neuroevolution in Low Dimensional Latent Space"
_ICLR.cc/2019/Conference_

### Official Review · AnonReviewer3 · 2018-11-02
**Contributions of this paper are unclear and not well evaluated**

**Rating:** 4
**Confidence:** 4

**Review:**

The main difficulty of neuroevolution---requiring a huge number of simulations for high dimensional problem---is addressed in this paper by introducing VAE to reduce the state space dimensionality and using a rather shallow controller network. This idea itself is very promising, however, it has been introduced in (Ha and Schmidhuber, 2018).  Still, there seems to be differences in how to gather histories and how to use them. Nevertheless, the differences are not well described in the text. The effect of the modification is not evaluated on experiments.

---

### Official Review · AnonReviewer1 · 2018-11-02
**Review for "Sample Efficient Deep Neuroevolution in Low Dimensional Latent Space"**

**Rating:** 5
**Confidence:** 5

**Review:**

Summary: They propose "Sample Efficient Deep Neuroevolution" (SEDN) model and experiment on Atari games. In this model, they use a Variational Encoder (VAE) to encode state frames into a latent vector, and use an LSTM to encode the current latent vector and action to predict the next latent space. A policy network (trained using CMA-ES) takes the latent space, and hidden state of the RNN as an input, and outputs an action to execute.

The strengths of this paper is that it is clearly written. They even explain details of RL, background of evolution strategies, motivation of using CMA-ES (and also the algorithm itself, which is no small ordeal), so it might a good background review paper of the literature. The experimental setup is relatively easy to understand. I suggest putting Algorithm 3 before Table 1 since presenting the algorithm before results may be more natural.

That being said, there are issues with this work that needs to be addressed before publication. I will list the issues and some suggestions I have, in order to help make this work better, hopefully good enough for acceptance:

1) The authors cited [1] a few times in the paper, but actually their approach of using a VAE to compress frames into a latent, an LSTM to predict the next latent, and a CMA-ES trained network for the policy is precisely what is proposed in [1] (which had experiments that trained on the actual environment, like in this work, and also the generated environment). This paper reads like they have proposed the setup, and lacks clarity as to which parts are their contribution, and which parts are prior work, which I believe is important for a paper submitted to an academic venue. Not to say at all that there's no contribution or originally in this work - there are many, but I feel they should list out which bits are their contribution, and which bits are prior work more clearly. Doing so will make this paper and their contributions stronger.

In my opinion, their contributions are: Expanding on the approach of [1] to study on a larger set of environments (the Atari suite), where they also incorporated an iterative training loop (described Algorithm 3) that was not used in [1]. Also, unlike [1], they used a multi-layer policy network, and also explained and rationalized the intuition behind the choice of CMA-ES. I think by listing out the contributions, and separating them from previous work, the paper will be much stronger.

2) The results are not terribly strong. They achieved good results on 7 games out of 50 using 10M frames. To me, that's actually not a deal breaker, since research is not a SOTA game, but I would like to see a more detailed analysis of why the algorithm works, and when it fails so people know what future work needs to be done to address this. I'm also not convinced that using CMA-ES would have an advantage over A3C (with the same latent / hidden features going into A3C as inputs), so perhaps the author might achieve better results if A3C was used to train the policy network (or not, but would be nice to see this experiment). It would offer more insight if we know what kind of terminal scores can be achieved using this algorithm, if it were allowed to train for 1B frames like the other 2 setup. Finally, if the author was able to show that training inside a generated environment, even for pre-training before going back to the actual environment, helps sample efficiency, that would be a very interesting result to me.

I'm assigning a preliminary score of 5 for this work for now, but if the author address point (1) to my satisfaction I will revise the score to +1 points, and if the author is able to achieve much better results, or address items in point (2) to my satisfaction, I will revise the score by +1 or potentially +2 points, so the final score of this work may lie in the range of 5 -> 8. I feel the author should be able to improve the paper to get a score of 6-7 in the end, at least from me. Good luck!

[1] https://arxiv.org/abs/1803.10122

---

### Official Review · AnonReviewer2 · 2018-11-03
**Interesting but not good enough for now**

**Rating:** 4
**Confidence:** 4

**Review:**

This paper proposes a combination of Evolutionary methods and variational representation learning to improve the sample efficiency of RL methods.
They train a VAE on environment frames, as well as an action-conditioned Dynamics model to predict the next frames, and these form the representations fed into a policy network which is trained through ES.

Overall, I find the problem setting interesting, and they try to tackle Atari games instead of simpler domains.
The use of CMA-ES instead of NES is a good improvement, and the way they motivate using VAE representations to obtain manageable representation sizes is well put forward.
[Edit: as mentioned by the other reviewers, this extension isn't as novel, given Ha et al's work, hence that reduces my confidence about accepting this paper further...]

However, this paper suffers from several issues in its current state:
1.	Its presentation is overly detailed about known literature. Section 2 goes in low-level details which are not necessary. It covers ES methods even though a citation to Salimans et al. 2017 would have been sufficient. Section 2.1 is a really complete coverage of CMA-ES, which should really just be a citation of the actual paper again or should be in the Appendix, this doesn’t warrant 1.5 pages of the main text.
2.	The actual model presentation is too succinct and split into Section 2.2 and Section 3 (network architectures and parameters). It is never clear how many parameters are optimized by CMA-ES (I counted ~8200 parameters if the MLP of size 256 x 32 x n_a is used). Algorithm 3 however was extremely clear and helpful to fully understand the method.
3.	There is no clear evaluation of the performance of the VAE representations and of the RNN dynamics model. Did they actually learn to represent anything at all? Figure 4 is not sufficient in providing evidence supporting this.  Compare this to Higgins et al. 2017, which used VAEs which represented enough information to perform at the same performance as non-variational representations.
4.	This feeds into the biggest issue with the current results: The proposed method works rather badly, obtaining worst performance than ES on 35 out of 51 games (68%). On most of these games, the proposed method does not seem to be able to get off the ground at all.  Why is that the case? Obviously if the VAE+RNN do not represent the games well enough, the performance will be bad. Did the policy learning with CMA-ES converge well? (seeing learning curves might help) The fact that no gradients are passed back from the Policy to the VAE/RNN clearly emphasises that issue (The policy only affect the data on which the representations are periodically retrained on).

In conclusion, even though I feel this paper tries to tackle an interesting problem, the results are not sufficient to support them as of now.

Typos:
-	“Donates” instead of “denotes” in a few places.

References:
-	Higgins et al., 2017: https://arxiv.org/abs/1707.08475

---

### Meta-Review · Area_Chair1 · 2018-12-14

**Confidence:** 5
**Recommendation:** Reject

**Metareview:**

Pros:
- compelling idea to use VAEs to reduce the dimensionality of the space in which to run evolution
- non-trivial benchmark results
- clearly written, solid background

Cons:
- moderate novelty (as compared to [1])
- performance results are sup-par
- no rebuttal, despite constructive and detailed review comments (and an explicit willingness to raise scores by multiple points!)

The reviewers agree that the paper should be rejected in its current form, but would plausibly have been willing to reassess their scores for a major revision -- which did not materialize.